# Local Maxima in the Likelihood of Gaussian Mixture Models: Structural Results and Algorithmic Consequences

**Chi Jin**
UC Berkeley
chijin@cs.berkeley.edu

**Yuchen Zhang**
UC Berkeley
yuczhang@berkeley.edu

**Sivaraman Balakrishnan**
Carnegie Mellon University
siva@stat.cmu.edu

**Martin J. Wainwright**
UC Berkeley
wainwrig@berkeley.edu

**Michael I. Jordan**
UC Berkeley
jordan@cs.berkeley.edu

## Abstract

We provide two fundamental results on the population (infinite-sample) likelihood function of Gaussian mixture models with $M \geq 3$ components. Our first main result shows that the population likelihood function has bad local maxima even in the special case of equally-weighted mixtures of well-separated and spherical Gaussians. We prove that the log-likelihood value of these bad local maxima can be arbitrarily worse than that of any global optimum, thereby resolving an open question of Srebro [2007]. Our second main result shows that the EM algorithm (or a first-order variant of it) with random initialization will converge to bad critical points with probability at least $1 - e^{-\Omega(M)}$. We further establish that a first-order variant of EM will not converge to strict saddle points almost surely, indicating that the poor performance of the first-order method can be attributed to the existence of bad local maxima rather than bad saddle points. Overall, our results highlight the necessity of careful initialization when using the EM algorithm in practice, even when applied in highly favorable settings.

## 1  Introduction

Finite mixture models are widely used in variety of statistical settings, as models for heterogeneous populations, as flexible models for multivariate density estimation and as models for clustering. Their ability to model data as arising from underlying subpopulations provides essential flexibility in a wide range of applications Titterington [1985]. This combinatorial structure also creates challenges for statistical and computational theory, and there are many problems associated with estimation of finite mixtures that are still open. These problems are often studied in the setting of Gaussian mixture models (GMMs), reflecting the wide use of GMMs in applications, particular in the multivariate setting, and this setting will also be our focus in the current paper.

Early work [Teicher, 1963] studied the identifiability of finite mixture models, and this problem has continued to attract significant interest (see the recent paper of Allman et al. [2009] for a recent overview). More recent theoretical work has focused on issues related to the use of GMMs for the density estimation problem [Genovese and Wasserman, 2000, Ghosal and Van Der Vaart, 2001]. Focusing on rates of convergence for parameter estimation in GMMs, Chen [1995] established the surprising result that when the number of mixture components is unknown, then the standard $\sqrt{n}$-rate for regular parametric models is not achievable. Recent investigations [Ho and Nguyen, 2015] into

exact-fitted, under-fitted and over-fitted GMMs have characterized the achievable rates of convergence in these settings.

From an algorithmic perspective, the dominant practical method for estimating GMMs is the Expectation-Maximization (EM) algorithm [Dempster et al., 1977]. The EM algorithm is an ascent method for maximizing the likelihood, but is only guaranteed to converge to a stationary point of the likelihood function. As such, there are no general guarantees for the quality of the estimate produced via the EM algorithm for Gaussian mixture models.[1] This has led researchers to explore various alternative algorithms which are computationally efficient, and for which rigorous statistical guarantees can be given. Broadly, these algorithms are based either on clustering [Arora et al., 2005, Dasgupta and Schulman, 2007, Vempala and Wang, 2002, Chaudhuri and Rao, 2008] or on the method of moments [Belkin and Sinha, 2010, Moitra and Valiant, 2010, Hsu and Kakade, 2013].

Although general guarantees have not yet emerged, there has nonetheless been substantial progress on the theoretical analysis of EM and its variations. Dasgupta and Schulman [2007] analyzed a two-round variant of EM, which involved over-fitting the mixture and then pruning extra centers. They showed that this algorithm can be used to estimate Gaussian mixture components whose means are separated by at least $\Omega(d^{1/4})$. Balakrishnan et al. [2015] studied the local convergence of the EM algorithm for a mixture of two Gaussians with $\Omega(1)$-separation. Their results show that global optima have relatively large regions of attraction, but still require that the EM algorithm be provided with a reasonable initialization in order to ensure convergence to a near globally optimal solution.

To date, computationally efficient algorithms for estimating a GMM provide guarantees under the strong assumption that the samples come from a mixture of Gaussians—i.e., that the model is well-specified. In practice however, we never expect the data to exactly follow the generative model, and it is important to understand the robustness of our algorithms to this assumption. In fact, maximum likelihood has favorable properties in this regard—maximum-likelihood estimates are well known to be robust to perturbations in the Kullback-Leibler metric of the generative model [Donoho and Liu, 1988]. This mathematical result motivates further study of EM and other likelihood-based methods from the computational point of view. It would be useful to characterize when efficient algorithms can be used to compute a maximum likelihood estimate, or a solution that is nearly as accurate, and which retains the robustness properties of the maximum likelihood estimate.

In this paper, we focus our attention on uniformly weighted mixtures of $M$ isotropic Gaussians. For this favorable setting, Srebro [2007] conjectured that any local maximum of the likelihood function is a global maximum in the limit of infinite samples—in other words, that there are no bad local maxima for the population GMM likelihood function. This conjecture, if true, would provide strong theoretical justification for EM, at least for large sample sizes. For suitably small sample sizes, it is known [Améndola et al., 2015] that configurations of the samples can be constructed which lead to the likelihood function having an unbounded number of local maxima. The conjecture of Srebro [2007] avoids this by requiring that the samples come from the specified GMM, as well as by considering the (infinite-sample-size) population setting. In the context of high-dimensional regression, it has been observed that in some cases despite having a non-convex objective function, every local optimum of the objective is within a small, vanishing distance of a global optimum [see, e.g., Loh and Wainwright, 2013, Wang et al., 2014]. In these settings, it is indeed the case that for sufficiently large sample sizes there are no bad local optima.

**A mixture of two spherical Gaussians:** A Gaussian mixture model with a single component is simply a Gaussian, so the conjecture of Srebro [2007] holds trivially in this case. The first interesting case is a Gaussian mixture with two components, for which empirical evidence supports the conjecture that there are no bad local optima. It is possible to visualize the setting when there are only two components and to develop a more detailed understanding of the population likelihood surface.

Consider for instance a one-dimensional equally weighted unit variance GMM with true centers $\mu_1^* = -4$ and $\mu_2^* = 4$, and consider the log-likelihood as a function of the vector $\boldsymbol{\mu} := (\mu_1, \mu_2)$. Figure 1 shows both the population log-likelihood, $\boldsymbol{\mu} \mapsto \mathcal{L}(\boldsymbol{\mu})$, and the negative 2-norm of its gradient, $\boldsymbol{\mu} \mapsto -\|\nabla \mathcal{L}(\boldsymbol{\mu})\|_2$. Observe that the only local maxima are the vectors $(-4, 4)$ and $(4, -4)$,

which are both also global maxima. The only remaining critical point is $(0, 0)$, which is a saddle point. Although points of the form $(0, R), (R, 0)$ have small gradient when $|R|$ is large, the gradient is not exactly zero for any finite $R$. Rigorously resolving the question of existence or non-existence of local maxima for the setting when $M = 2$ remains an open problem.

In the remainder of our paper, we focus our attention on the setting where there are more than two mixture components and attempt to develop a broader understanding of likelihood surfaces for these models, as well as the consequences for algorithms.

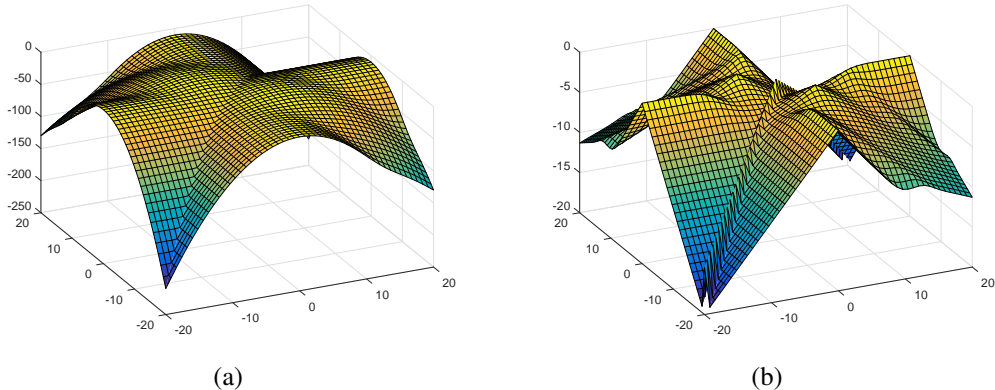

|     |     |
| :-: | :-: |
| (a) | (b) |

Figure 1: Illustration of the likelihood and gradient maps for a two-component Gaussian mixture. (a) Plot of population log-likelihood map $\boldsymbol{\mu} \mapsto \mathcal{L}(\boldsymbol{\mu})$. (b) Plot of the negative Euclidean norm of the gradient map $\boldsymbol{\mu} \mapsto -\|\nabla \mathcal{L}(\boldsymbol{\mu})\|_2$.

Our first contribution is a negative answer to the open question of Srebro [2007]. We construct a GMM which is a uniform mixture of three spherical unit variance, well-separated, Gaussians whose population log-likelihood function contains local maxima. We further show that the log-likelihood of these local maxima can be arbitrarily worse than that of the global maxima. This result immediately implies that any local search algorithm cannot exhibit global convergence (meaning convergence to a global optimum from all possible starting points), even on well-separated mixtures of Gaussians.

The mere existence of bad local maxima is not a practical concern unless it turns out that natural algorithms are frequently trapped in these bad local maxima. Our second main result shows that the EM algorithm, as well as a variant thereof known as the first-order EM algorithm, with random initialization, converges to a bad critical point with an exponentially high probability. In more detail, we consider the following practical scheme for parameter estimation in an $M$-component Gaussian mixture:

(a) Draw $M$ i.i.d. points $\mu_1, \ldots, \mu_M$ uniformly at random from the sample set.

(b) Run the EM or first-order EM algorithm to estimate the model parameters, using $\mu_1, \ldots, \mu_M$ as the initial centers.

We note that in the limit of infinite samples, the initialization scheme we consider is equivalent to selecting $M$ initial centers i.i.d from the underlying mixture distribution. We show that for a universal constant $c > 0$, with probability at least $1 - e^{-cM}$, the EM and first-order EM algorithms converge to a suboptimal critical point, whose log-likelihood could be arbitrarily worse than that of the global maximum. Conversely, in order to find a solution with satisfactory log-likelihood via this initialization scheme, one needs repeat the above scheme exponentially many (in $M$) times, and then select the solution with highest log-likelihood. This result strongly indicates that repeated random initialization followed by local search (via either EM or its first order variant) can fail to produce useful estimates under reasonable constraints on computational complexity.

We further prove that under the same random initialization scheme, the first-order EM algorithm with a suitable stepsize *does not* converge to a strict saddle point with probability one. This fact strongly suggests that the failure of local search methods for the GMM model is due mainly to the existence of bad local optima, and not due to the presence of (strict) saddle points.

Our proofs introduce new techniques to reason about the structure of the population log-likelihood, and in particular to show the existence of bad local optima. We expect that these general ideas will aid in developing a better understanding of the behavior of algorithms for non-convex optimization. From a practical standpoint, our results strongly suggest that careful initialization is required for local search methods, even in large-sample settings, and even for extremely well-behaved mixture models.

The remainder of this paper is organized as follows. In Section 2, we introduce GMMs, the EM algorithm, its first-order variant and we formally set up the problem we consider. In Section 3, we state our main theoretical results and develop some of their implications. Section A is devoted to the proofs of our results, with some of the more technical aspects deferred to the appendices.

## 2 Background and Preliminaries

In this section, we formally define the Gaussian mixture model that we study in the paper. We then describe the EM algorithm, the first-order EM algorithm, as well as the form of random initialization that we analyze. Throughout the paper, we use $[M]$ to denote the set $\{1, 2, \cdots, M\}$, and $\mathcal{N}(\mu, \Sigma)$ to denote the $d$-dimensional Gaussian distribution with mean vector $\mu$ and covariance matrix $\Sigma$. We use $\phi(\cdot \mid \mu, \Sigma)$ to denote the probability density function of the Gaussian distribution with mean vector $\mu$ and covariance matrix $\Sigma$:

$$\phi(x \mid \mu, \Sigma) := \frac{1}{\sqrt{(2\pi)^d \det(\Sigma)}} e^{-\frac{1}{2}(x-\mu)^\top \Sigma^{-1}(x-\mu)}. \tag{1}$$

### 2.1 Gaussian Mixture Models

A $d$-dimensional Gaussian mixture model (GMM) with $M$ components can be specified by a collection $\boldsymbol{\mu}^* = \{\mu_i^*, \ldots, \mu_M^*\}$ of $d$-dimensional mean vectors, a vector $\boldsymbol{\lambda}^* = (\lambda_1^*, \ldots, \lambda_M^*)$ of non-negative mixture weights that sum to one, and a collection $\boldsymbol{\Sigma}^* = \{\Sigma_1^*, \ldots, \Sigma_M^*\}$ of covariance matrices. Given these parameters, the density function of a Gaussian mixture model takes the form

$$p(x \mid \boldsymbol{\lambda}^*, \boldsymbol{\mu}^*, \boldsymbol{\Sigma}^*) = \sum_{i=1}^{M} \lambda_i^* \phi(x \mid \mu_i^*, \Sigma_i^*),$$

where the Gaussian density function $\phi$ was previously defined in equation (1). In this paper, we focus on the idealized situation in which every mixture component is equally weighted, and the covariance of each mixture component is the identity. This leads to a mixture model of the form

$$p(x \mid \boldsymbol{\mu}^*) := \frac{1}{M} \sum_{i=1}^{M} \phi(x \mid \mu_i^*, \mathrm{I}), \tag{2}$$

which we denote by $\mathrm{GMM}(\boldsymbol{\mu}^*)$. In this case, the only parameters to be estimated are the mean vectors $\boldsymbol{\mu}^* = \{\mu_i^*\}_{i=1}^{M}$ of the $M$ components.

The difficulty of estimating a Gaussian mixture distribution depends on the amount of separation between the mean vectors. More precisely, for a given parameter $\xi > 0$, we say that the $\mathrm{GMM}(\boldsymbol{\mu}^*)$-model is $\xi$-*separated* if

$$\|\mu_i^* - \mu_j^*\|_2 \geq \xi, \quad \text{for all distinct pairs } i, j \in [M]. \tag{3}$$

We say that the mixture is *well-separated* if condition (3) holds for some $\xi = \Omega(\sqrt{d})$.

Suppose that we observe an i.i.d. sequence $\{x_\ell\}_{\ell=1}^{n}$ drawn according to the distribution $\mathrm{GMM}(\boldsymbol{\mu}^*)$, and our goal is to estimate the unknown collection of mean vectors $\boldsymbol{\mu}^*$. The sample-based log-likelihood function $\mathcal{L}_n$ is given by

$$\mathcal{L}_n(\boldsymbol{\mu}) := \frac{1}{n} \sum_{\ell=1}^{n} \log \left( \frac{1}{M} \sum_{i=1}^{M} \phi(x_\ell \mid \mu_i, \mathrm{I}) \right). \tag{4a}$$

As the sample size $n$ tends to infinity, this sample likelihood converges to the population log-likelihood function $\mathcal{L}$ given by

$$\mathcal{L}(\boldsymbol{\mu}) = \mathbb{E}_{\boldsymbol{\mu}^*} \log \left( \frac{1}{M} \sum_{i=1}^{M} \phi(X \mid \mu_i, \mathrm{I}) \right). \tag{4b}$$

Here $\mathbb{E}_{\boldsymbol{\mu}^*}$ denotes expectation taken over the random vector $X$ drawn according to the model GMM($\boldsymbol{\mu}^*$).

A straightforward implication of the positivity of the KL divergence is that the population likelihood function is in fact maximized at $\boldsymbol{\mu}^*$ (along with permutations thereof, depending on how we index the mixture components). On the basis of empirical evidence, Srebro [2007] conjectured that this population log-likelihood is in fact well-behaved, in the sense of having no spurious local optima. In Theorem 1, we show that this intuition is false, and provide a simple example of a mixture of $M = 3$ well-separated Gaussians in dimension $d = 1$, whose population log-likelihood function has arbitrarily bad local optima.

## 2.2 Expectation-Maximization Algorithm

A natural way to estimate the mean vectors $\boldsymbol{\mu}^*$ is by attempting to maximize the sample log-likelihood defined by the samples $\{x_\ell\}_{\ell=1}^n$. For a non-degenerate Gaussian mixture model, the log-likelihood is non-concave. Rather than attempting to maximize the log-likelihood directly, the EM algorithm proceeds by iteratively maximizing a lower bound on the log-likelihood. It does so by alternating between two steps:

1. E-step: For each $i \in [M]$ and $\ell \in [n]$, compute the *membership weight*

$$w_i(x_\ell) = \frac{\phi(x_\ell \mid \mu_i, \mathrm{I})}{\sum_{j=1}^M \phi(x_\ell \mid \mu_j, \mathrm{I})}.$$

2. M-step: For each $i \in [M]$, update the mean $\mu_i$ vector via

$$\mu_i^{\mathrm{new}} = \frac{\sum_{i=1}^n w_i(x_\ell)\, x_\ell}{\sum_{\ell=1}^n w_i(x_\ell)}.$$

In the population setting, the M-step becomes:

$$\mu_i^{\mathrm{new}} = \frac{\mathbb{E}_{\boldsymbol{\mu}^*}[w_i(X)\, X]}{\mathbb{E}_{\boldsymbol{\mu}^*}[w_i(X)]}. \tag{5}$$

Intuitively, the M-step updates the mean vector of each Gaussian component to be a weighted centroid of the samples for appropriately chosen weights.

**First-order EM updates:** For a general latent variable model with observed variables $X = x$, latent variables $Z$ and model parameters $\theta$, by Jensen's inequality, the log-likelihood function can be lower bounded as

$$\log \mathbb{P}(x \mid \theta') \geq \underbrace{\mathbb{E}_{Z \sim \mathbb{P}(\cdot \mid x;\theta)} \log \mathbb{P}(x, Z \mid \theta')}_{:=Q(\theta' \mid \theta)} - \mathbb{E}_{Z \sim \mathbb{P}(\cdot \mid x;\theta)} \log \mathbb{P}(Z \mid x; \theta').$$

Each step of the EM algorithm can also be viewed as optimizing over this lower bound, which gives:

$$\theta^{\mathrm{new}} := \arg\max_{\theta'} Q(\theta' \mid \theta)$$

There are many variants of the EM algorithm which rely instead on partial updates at each iteration instead of finding the exact optimum of $Q(\theta' \mid \theta)$. One important example, analyzed in the work of Balakrishnan et al. [2015], is the first-order EM algorithm. The first-order EM algorithm takes a step along the gradient of the function $Q(\theta' \mid \theta)$ (with respect to its first argument) in each iteration. Concretely, given a step size $s > 0$, the first-order EM updates can be written as:

$$\theta^{\mathrm{new}} = \theta + s \nabla_{\theta'} Q(\theta' \mid \theta) \mid_{\theta'=\theta}.$$

In the case of the model GMM($\mu^*$), the gradient EM updates on the population objective take the form

$$\mu_i^{\mathrm{new}} = \mu_i + s\, \mathbb{E}_{\boldsymbol{\mu}^*}\big[w_i(X)(X - \mu_i)\big]. \tag{6}$$

This update turns out to be equivalent to gradient ascent on the population likelihood $\mathcal{L}$ with step size $s > 0$ (see the paper Balakrishnan et al. [2015] for details).

## 2.3 Random Initialization

Since the log-likelihood function is non-concave, the point to which the EM algorithm converges depends on the initial value of $\boldsymbol{\mu}$. In practice, it is standard to choose these values by some form of random initialization. For instance, one method is to to initialize the mean vectors by sampling uniformly at random from the data set $\{x_\ell\}_{\ell=1}^n$. This scheme is intuitively reasonable, because it automatically adapts to the locations of the true centers. If the true centers have large mutual distances, then the initialized centers will also be scattered. Conversely, if the true centers concentrate in a small region of the space, then the initialized centers will also be close to each other. In practice, initializing $\boldsymbol{\mu}$ by uniformly drawing from the data is often more reasonable than drawing $\boldsymbol{\mu}$ from a fixed distribution.

In this paper, we analyze the EM algorithm and its variants at the population level. We focus on the above practical initialization scheme of selecting $\boldsymbol{\mu}$ uniformly at random from the sample set. In the idealized population setting, this is equivalent to sampling the initial values of $\boldsymbol{\mu}$ i.i.d from the distribution GMM($\boldsymbol{\mu}^*$). Throughout this paper, we refer to this particular initialization strategy as *random initialization.*

## 3 Main results

We now turn to the statements of our main results, along with a discussion of some of their consequences.

### 3.1 Structural properties

In our first main result (Theorem 1), for any $M \geq 3$, we exhibit an $M$-component mixture of Gaussians in dimension $d = 1$ for which the population log-likelihood has a bad local maximum whose log-likelihood is arbitrarily worse than that attained by the true parameters $\boldsymbol{\mu}^*$. This result provides a negative answer to the conjecture of Srebro [2007].

**Theorem 1.** *For any $M \geq 3$ and any constant $C_{\mathrm{gap}} > 0$, there is a well-separated uniform mixture of $M$ unit-variance spherical Gaussians GMM($\boldsymbol{\mu}^*$) and a local maximum $\boldsymbol{\mu}'$ such that*

$$\mathcal{L}(\boldsymbol{\mu}') \leq \mathcal{L}(\boldsymbol{\mu}^*) - C_{\mathrm{gap}}.$$

In order to illustrate the intuition underlying Theorem 1, we give a geometrical description of our construction for $M = 3$. Suppose that the true centers $\mu_1^*$, $\mu_2^*$ and $\mu_3^*$, are such that the distance between $\mu_1^*$ and $\mu_2^*$ is much smaller than the respective distances from $\mu_1^*$ to $\mu_3^*$, and from $\mu_2^*$ to $\mu_3^*$. Now, consider the point $\boldsymbol{\mu} := (\mu_1, \mu_2, \mu_3)$ where $\mu_1 = (\mu_1^* + \mu_2^*)/2$; the points $\mu_2$ and $\mu_3$ are both placed at the true center $\mu_3^*$. This assignment does not maximize the population log-likelihood, because only one center is assigned to the two Gaussian components centered at $\mu_1^*$ and $\mu_2^*$, while two centers are assigned to the Gaussian component centered at $\mu_3^*$. However, when the components are well-separated we are able to show that there is a local maximum in the neighborhood of this configuration. In order to establish the existence of a local maximum, we first define a neighborhood of this configuration ensuring that it does not contain any global maximum, and then prove that the log-likelihood on the boundary of this neighborhood is strictly smaller than that of the sub-optimal configuration $\boldsymbol{\mu}$. Since the log-likelihood is bounded from above, this neighborhood must contain at least one maximum of the log-likelihood. Since the global maxima are not in this neighborhood by construction, any maximum in this neighborhood must be a local maximum. See Section A for a detailed proof.

### 3.2 Algorithmic consequences

An important implication of Theorem 1 is that any iterative algorithm, such as EM or gradient ascent, that attempts to maximize the likelihood based on local updates *cannot* be globally convergent—that is, cannot converge to (near) globally optimal solutions from an arbitrary initialization. Indeed, if any such algorithm is initialized at the local maximum, then they will remain trapped. However, one might argue that this conclusion is overly pessimistic, in that we have only shown that these algorithms fail when initialized at a certain (adversarially chosen) point. Indeed, the mere existence of bad local minima need not be a practical concern unless it can be shown that a typical optimization

algorithm will frequently converge to one of them. The following result shows that the EM algorithm, when applied to the population likelihood and initialized according to the random scheme described in Section 2.2, converges to a bad critical point with high probability.

**Theorem 2.** *Let $\boldsymbol{\mu}^t$ be the $t^{\text{th}}$ iterate of the EM algorithm initialized by the random initialization scheme described previously. There exists a universal constant $c$, for any $M \geq 3$ and any constant $C_{\text{gap}} > 0$, such that there is a well-separated uniform mixture of $M$ unit-variance spherical Gaussians $GMM(\boldsymbol{\mu}^*)$ with*

$$\mathbb{P}\left[\forall t \geq 0,\ \mathcal{L}(\boldsymbol{\mu}^t) \leq \mathcal{L}(\boldsymbol{\mu}^*) - C_{\text{gap}}\right] \geq 1 - e^{-cM}.$$

Theorem 2 shows that, for the specified configuration $\boldsymbol{\mu}^*$, the probability of success for the EM algorithm is exponentially small as a function of $M$. As a consequence, in order to guarantee recovering a global maximum with at least constant probability, the EM algorithm with random initialization must be executed at least $e^{\Omega(M)}$ times. This result strongly suggests that that effective initialization schemes, such as those based on pilot estimators utilizing the method of moments [Moitra and Valiant, 2010, Hsu and Kakade, 2013], are critical to finding good maxima in general GMMs.

The key idea in the proof of Theorem 2 is the following: suppose that all the true centers are grouped into two clusters that are extremely far apart, and suppose further that we initialize all the centers in the neighborhood of these two clusters, while ensuring that at least one center lies within each cluster. In this situation, all centers will remain trapped within the cluster in which they were first initialized, irrespective of how many steps we take in the EM algorithm. Intuitively, this suggests that the only favorable initialization schemes (from which convergence to a global maximum is possible) are those in which (1) all initialized centers fall in the neighborhood of exactly one cluster of true centers, (2) the number of centers initialized within each cluster of true centers exactly matches the number of true centers in that cluster. However, this observation alone only suffices to guarantee that the success probability is polynomially small in $M$.

In order to demonstrate that the success probability is *exponentially small* in $M$, we need to further refine this construction. In more detail, we construct a Gaussian mixture distribution with a recursive structure: on top level, its true centers can be grouped into two clusters far apart, and then inside each cluster, the true centers can be further grouped into two mini-clusters which are well-separated, and so on. We can repeat this structure for $\Omega(\log M)$ levels. For this GMM instance, even in the case where the number of true centers exactly matches the number of initialized centers in each cluster at the top level, we still need to consider the configuration of the initial centers within the mini-clusters, which further reduces the probability of success for a random initialization. A straightforward calculation then shows that the probability of a favorable random initialization is on the order of $e^{-\Omega(M)}$. The full proof is given in Section A.2.

We devote the remainder of this section to a treatment of the first-order EM algorithm. Our first result in this direction shows that the problem of convergence to sub-optimal fixed points remains a problem for the first-order EM algorithm, provided the step-size is not chosen too aggressively.

**Theorem 3.** *Let $\boldsymbol{\mu}^t$ be the $t^{\text{th}}$ iterate of the first-order EM algorithm with stepsize $s \in (0,1)$, initialized by the random initialization scheme described previously. There exists a universal constant $c$, for any $M \geq 3$ and any constant $C_{\text{gap}} > 0$, such that there is a well-separated uniform mixture of $M$ unit-variance spherical Gaussians $GMM(\boldsymbol{\mu}^*)$ with*

$$\mathbb{P}\left(\forall t \geq 0,\ \mathcal{L}(\boldsymbol{\mu}^t) \leq \mathcal{L}(\boldsymbol{\mu}^*) - C_{\text{gap}}\right) \geq 1 - e^{-cM}. \tag{7}$$

We note that the restriction on the step-size is weak, and is satisfied by the theoretically optimal choice for a mixture of two Gaussians in the setting studied by Balakrishnan et al. [2015]. Recall that the first-order EM updates are identical to gradient ascent updates on the log-likelihood function. As a consequence, we can conclude that the most natural local search heuristics for maximizing the log-likelihood (EM and gradient ascent), fail to provide statistically meaningful estimates when initialized randomly, unless we repeat this procedure exponentially many (in $M$) times.

Our final result concerns the type of fixed points reached by the first-order EM algorithm in our setting. Pascanu et al. [2014] argue that for high-dimensional optimization problems, the principal difficulty is the proliferation of saddle points, not the existence of poor local maxima. In our setting, however, we can leverage recent results on gradient methods [Lee et al., 2016, Panageas and Piliouras, 2016] to show that the first-order EM algorithm cannot converge to strict saddle points. More precisely:

**Definition 1** (Strict saddle point Ge et al. [2015]). *For a maximization problem, we say that a critical point $\mathbf{x}_{ss}$ of function $f$ is a* strict saddle point *if the Hessian $\nabla^2 f(\mathbf{x}_{ss})$ has at least one strictly positive eigenvalue.*

With this definition, we have the following:

**Theorem 4.** *Let $\boldsymbol{\mu}^t$ be the $t^{\text{th}}$ iterate of the first-order EM algorithm with constant stepsize $s \in (0,1)$, and initialized by the random initialization scheme described previously. Then for any $M$-component mixture of spherical Gaussians:*

(a) *The iterates $\boldsymbol{\mu}^t$ converge to a critical point of the log-likelihood.*

(b) *For any strict saddle point $\boldsymbol{\mu}_{ss}$, we have $\mathbb{P}\left(\lim_{t\to\infty}\boldsymbol{\mu}^t = \boldsymbol{\mu}_{ss}\right) = 0$.*

Theorems 3 and 4 provide strong support for the claim that the sub-optimal points to which the first-order EM algorithm frequently converges are bad local maxima. The algorithmic failure of the first-order EM algorithm is most likely due to the presence of bad local maxima, as opposed to (strict) saddle-points.

The proof of Theorem 4 is based on recent work [Lee et al., 2016, Panageas and Piliouras, 2016] on the asymptotic performance of gradient methods. That work reposes on the stable manifold theorem from dynamical systems theory, and, applied directly to our setting, would require establishing that the population likelihood $\mathcal{L}$ is smooth. Our proof technique avoids such a smoothness argument; see Section A.4 for the details. The proof technique makes use of specific properties of the first-order EM algorithm that do not hold for the EM algorithm. We conjecture that a similar result is true for the EM algorithm; however, we suspect that a generalized version of the stable manifold theorem will be needed to establish such a result.

## 4 Conclusion and open problems

In this paper, we resolved an open problem of Srebro [2007], by demonstrating the existence of arbitrarily bad local maxima for the population log-likelihood of Gaussian mixture model, even in the idealized situation where each component is uniformly weighted, spherical with unit variance, and well-separated. We further provided some evidence that even in this favorable setting random initialization schemes for the population EM algorithm are likely to fail with high probability. Our results carry over in a straightforward way, via standard empirical process arguments, to settings where a large finite sample is provided.

An interesting open question is to resolve the necessity of at least three mixture components in our constructions. In particular, we believe that at least three mixture components are necessary for the log-likelihood to be poorly behaved, and that for a well-separated mixture of two Gaussians the EM algorithm with a random initialization is in fact successful with high probability.

In a related vein, understanding the empirical success of EM-style algorithms using random initialization schemes despite their failure on seemingly benign problem instances remains an open problem which we hope to address in future work.

**Acknowledgements**

This work was partially supported by Office of Naval Research MURI grant DOD-002888, Air Force Office of Scientific Research Grant AFOSR-FA9550-14-1-001, the Mathematical Data Science program of the Office of Naval Research under grant number N00014-15-1-2670, and National Science Foundation Grant CIF-31712-23800.

## Footnotes

[1]In addition to issues of convergence to non-maximal stationary points, solutions of infinite likelihood exist for GMMs where both the location and scale parameters are estimated. In practice, several methods exist to avoid such solutions. In this paper, we avoid this issue by focusing on GMMs in which the scale parameters are fixed.

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
