[Supplementary Material]

# A    Proofs of Main Theorems

This section is devoted to the proofs of Theorems 1 through 4. Certain technical aspects of the proofs are deferred to later sections in appendix.

## A.1    Proof of Theorem 1

In this section, we prove Theorem 1. The proof consists of three parts: starting with the case $M = 3$, the first part shows the existence of a local maximum for certain GMMs, whereas the second part shows that this local maximum has a log-likelihood that is much worse than that of the global maximum. The third part provides the extension to the general case of $M > 3$ mixture components.

### A.1.1    Existence of a local maximum

In this section, we prove the existence of a local maximum by first constructing a family of GMMs parametrized by a scalar $\gamma$, and then proving the existence of local maxima in the limiting case when $\gamma \to +\infty$. By continuity of the log-likelihood function, we can then conclude that there exists some finite $\gamma$ whose corresponding log-likelihood has local maxima.

We begin by considering the special case of $M = 3$ components in dimension $d = 1$. For parameters $R > 0$ and $\gamma \gg 1$, suppose that the true centers $\boldsymbol{\mu}^*$ are given by

$$\mu_1^* = -R, \qquad \mu_2^* = R, \qquad \mu_3^* = \gamma R.$$

By construction, the two centers $\mu_1^*$ and $\mu_2^*$ are relatively close together near the origin, while the third center $\mu_3^*$ is located far away from both of the first two centers.

We first claim that when $\gamma$ is sufficiently large, there is a local maximum in the closed set:

$$\mathcal{D} = \left\{ (\mu_1, \mu_2, \mu_3) \in \mathbb{R}^3 \ \middle| \ \mu_1 \le \frac{\gamma R}{3}, \ \mu_2 \ge \frac{2\gamma R}{3} \ \text{and} \ \mu_3 \ge \frac{2\gamma R}{3} \right\}.$$

To establish this claim, we consider the value of population log-likelihood function $\mathcal{L}(\tilde{\boldsymbol{\mu}})$ at an interior point $\tilde{\boldsymbol{\mu}} = (0, \gamma R, \gamma R)$ of $\mathcal{D}$, and compare it to the log-likelihood on the boundary of the set $\mathcal{D}$. We show that for a sufficiently large $\gamma$, the log-likelihood at the interior point is strictly larger than the log-likelihood on the boundary, and use this to argue that there must be a local maxima in the set $\mathcal{D}$. Concretely, define $v_0 := \mathcal{L}(\tilde{\boldsymbol{\mu}})$, and the maximum value of $\mathcal{L}(\boldsymbol{\mu})$ on the three two-dimensional faces of $\mathcal{D}$, i.e.,

$$v_1 := \sup_{\substack{\mu_1 = \gamma R/3 \\ \mu_2 \ge 2\gamma R/3 \\ \mu_3 \ge 2\gamma R/3}} \mathcal{L}(\boldsymbol{\mu}), \qquad v_2 := \sup_{\substack{\mu_1 \le \gamma R/3 \\ \mu_2 = 2\gamma R/3 \\ \mu_3 \ge 2\gamma R/3}} \mathcal{L}(\boldsymbol{\mu}), \qquad \text{and} \qquad v_3 := \sup_{\substack{\mu_1 \le \gamma R/3 \\ \mu_2 \ge 2\gamma R/3 \\ \mu_3 = 2\gamma R/3}} \mathcal{L}(\boldsymbol{\mu}).$$

The population log-likelihood function is given by the expression

$$\mathcal{L}(\boldsymbol{\mu}) = \mathbb{E}_{\boldsymbol{\mu}^*} \log \left( \sum_{i=1}^{3} e^{-\frac{1}{2}(X - \mu_i)^2} \right) - \log(3\sqrt{2\pi}).$$

As $\gamma \to \infty$, it is easy to verify that

$$v_0 = \mathcal{L}(\tilde{\boldsymbol{\mu}}) \to -\frac{2R^2 + 3 - 2\log(2)}{6} - \log(3\sqrt{2\pi}).$$

Similarly, we can calculate the value of $v_1, v_2$ and $v_3$ as $\gamma \to \infty$; i.e., a straightforward calculation shows that

$$\lim_{\gamma \to +\infty} v_1 = -\infty,$$

$$\lim_{\gamma \to +\infty} v_2 = -\frac{2R^2 + 3}{6} - \log(3\sqrt{2\pi}) \quad \text{(the maximum is attained at } \mu_1 \to 0 \text{ and } \mu_3 \to \gamma R),$$

$$\lim_{\gamma \to +\infty} v_3 = -\frac{2R^2 + 3}{6} - \log(3\sqrt{2\pi}) \quad \text{(the maximum is attained at } \mu_1 \to 0 \text{ and } \mu_2 \to \gamma R).$$

This gives the relation $v_0 > \max\{v_1, v_2, v_3\}$ when $\gamma \to \infty$. Since $\mathcal{L}$ is a continuous function of $\gamma$, we know that $v_0, v_1, v_2, v_3$ are also continuous functions of $\gamma$. Therefore, there exists a finite $A$ such that, as long as $\gamma > A$, we will still have $v_0 > \max\{v_1, v_2, v_3\}$. This in turn implies that the function value at an interior point is strictly greater than the function value on the boundary of $\mathcal{D}$, which implies the existence of at least one local maximum inside $\mathcal{D}$.

On the other hand, the global maxima of the population likelihood function are $(-R, R, \gamma R)$ and its permutations, which are not in $\mathcal{D}$. This shows the existence of at least one local maximum which is not a global maximum.

### A.1.2 Log-likelihood at a local maximum

In order to prove that the log-likelihood of a local maximum can be arbitrarily worse than the log-likelihood of the global maximum, we consider the limit when $R \to \infty$. In this case, the limiting value of the global maximum will be

$$\lim_{R \to \infty} \mathcal{L}(\boldsymbol{\mu}^*) = -\frac{1}{2} - \log(3\sqrt{2\pi}).$$

Let $\boldsymbol{\mu}' = (\mu_1', \mu_2', \mu_3')$ be one of the local maxima in the closed set $\mathcal{D}$. We have previously established the existence of such a local maximum.

Since $\mu_1^* - \mu_2^* = 2R$, we know that either $|\mu_2^* - \mu_1'| > R$ or $|\mu_1^* - \mu_1'| > R$ has to be true. Without loss of generality, we may assume that $|\mu_2^* - \mu_1'| > R$. From the definition of the set $\mathcal{D}$, we can also see that $|\mu_2^* - \mu_2'| > R$ and $|\mu_2^* - \mu_3'| > R$. Putting together the pieces yields

$$\lim_{R \to +\infty} \mathcal{L}(\boldsymbol{\mu}') \leq \lim_{R \to +\infty} \frac{1}{3} \mathbb{E}_{X \sim \mathcal{N}(\mu_2^*, 1)} \log\left(\sum_{i=1}^{3} e^{-\frac{1}{2}(X - \mu_i')^2}\right) - \frac{1}{3}\log(3\sqrt{2\pi}) = -\infty.$$

Again, by the continuity of the function $\mathcal{L}$ with respect to $R$, we know for any $C_{\text{gap}} > 0$, there always exists a large constant $A'$, so that if $R > A'$, we will have $\mathcal{L}(\boldsymbol{\mu}^*) - \mathcal{L}(\boldsymbol{\mu}') > C_{\text{gap}}$. This completes the proof for case $M = 3$.

### A.1.3 Extension to the case $M > 3$

We now provide an outline of how this argument can be extended to the general setting of $M > 3$. Consider a GMM with true centers

$$\mu_i^* = \frac{(2i - k)R}{k - 2}, \qquad \text{for } i = 1, \cdots, M - 1 \quad \text{and} \qquad \mu_M^* = \gamma R,$$

for some parameter $\gamma > 0$ to be chosen. We claim that when $\gamma$ is sufficiently large, there is at least one local maximum in the closed set

$$\mathcal{D}_M = \left\{ (\mu_1, \cdots, \mu_M) \mid \mu_1 \leq \frac{\gamma R}{3}, \ \mu_2 \geq \frac{2\gamma R}{3}, \cdots, \mu_M \geq \frac{2\gamma R}{3} \right\}.$$

The proof follows from an identical argument as in the $M = 3$ case.

## A.2 Proof of Theorem 2

In this section, we prove Theorem 2. We first present an important technical lemma that addresses the behavior of the EM algorithm for a particular configuration of true and initial centers. We then prove the theorem by constructing a bad example and recursively applying this lemma. The proof of this lemma is given in Appendix B.

We focus on the one-dimensional setting throughout this proof. We use $\mathbb{B}_x(\delta)$ to denote an interval centered at $x$ with radius $\delta$, that is, $\mathbb{B}_x(\delta) = [x - \delta, x + \delta]$. We also use $\overline{\mathbb{B}_x(\delta)}$ to represent the complement of the interval $\mathbb{B}_x(\delta)$, i.e. $\overline{\mathbb{B}_x(\delta)} = (-\infty, x - \delta) \cup (x + \delta, \infty)$.

As a preliminary, let us define a class of GMMs, which we refer to as *diffuse GMMs*. We say that a mixture model $\text{GMM}(\boldsymbol{\mu}^*)$ consisting of $\widetilde{M}$ components is $(c, \delta)$-*diffuse* if:

    (a) For some $M \leq \widetilde{M}$, there are $M$ centers contained in $\mathbb{B}_{c\delta}(\delta) \cup \mathbb{B}_{-c\delta}(\delta)$;

(b) Each of the sets $\mathbb{B}_{c\delta}(\delta)$ and $\mathbb{B}_{-c\delta}(\delta)$ contain at least one center;

(c) The remaining $\widetilde{M} - M$ centers are all in $\overline{\mathbb{B}_0(20c\delta)}$.

Consider the EM algorithm, and denote by $M_1^{(t)}, M_2^{(t)}$ and $M_3^{(t)}$ the number of centers the EM algorithm has in the $\mathrm{t^{th}}$ iteration in the sets $\mathbb{B}_{-c\delta}(2\delta), \mathbb{B}_{c\delta}(2\delta)$ and $\overline{\mathbb{B}_0(20c\delta)}$ respectively, where $c$ and $\delta$ are those specified in the definition of the diffuse GMM. To be clear, $M_1^{(0)}, M_2^{(0)}$ and $M_3^{(0)}$ denote the number of centers in these sets in the initial configuration specified to the EM algorithm. With these definitions in place, we can state our lemma.

**Lemma 1.** *Suppose that the true underlying distribution is a $(c, \delta)$-diffuse GMM with $c > 20$ and $\delta > \log M + 3$, and that the EM algorithm is initialized so that $M_1^{(0)}, M_2^{(0)} \geq 1$.*

*(a) If $M = \widetilde{M}$, then*

$$M_1^{(t)} = M_1^{(0)} \quad and \quad M_2^{(t)} = M_2^{(0)} \qquad for\ every\ t \geq 0. \tag{8}$$

*(b) If $M < \widetilde{M}$, suppose further that for each center in $\mu_j^* \in \overline{\mathbb{B}_0(20c\delta)}$, there is an initial center $\mu_{j'}^{(0)}$ such that $|\mu_{j'}^{(0)} - \mu_j^*| \leq |\mu_j^*|/10$. Then the same conditions (8) hold.*

Intuitively, these results show that if the true centers are clustered together into two clusters that are well separated, and the EM algorithm is initialized so that each cluster is accounted for by at least one initial center then the EM algorithm remains trapped in the initial configuration of centers. A concrete implication of part (a) is that if the true distribution is a $(c, \delta)$-diffuse GMM with $\widetilde{M} = M$ and $M_1^*, M_2^*$ true clusters lie in $\mathbb{B}_{-c\delta}(\delta)$ and $\mathbb{B}_{c\delta}(\delta)$ respectively, then there are only three possible ways to initialize the EM algorithm that might possibly converge to a global maximum of the log-likelihood function; i.e., the pair $(M_1^{(0)}, M_2^{(0)})$ must be one of $\{(M, 0), (M_1^*, M_2^*), (0, M)\}$, where $M = M_1^* + M_2^*$.

We are now equipped to prove Theorem 2. We will first focus on the case $M = 2^m$ for some positive integer $m$; the case of arbitrary $M$ will be addressed later. At a high level, we will first construct the distribution $\mathrm{GMM}(\boldsymbol{\mu}^*)$ that establishes the theorem, and then use the above technical lemma in order to reason about the behavior of the EM algorithm on this distribution.

**Case $M = 2^m$:** First, define the collection of $2^m$ binary vectors of the form $\boldsymbol{\epsilon} = (\epsilon_1, \epsilon_2, \cdots, \epsilon_m)$ where each $\epsilon_i \in \{-1, 1\}$. Consider the distribution, $\mathrm{GMM}(\boldsymbol{\mu}^*)$, with the locations of the true centers indexed by these $2^m$ vectors; i.e., each center is located at

$$\mu(\boldsymbol{\epsilon}) = \sum_{i=1}^{m} \epsilon_i \left(\frac{1}{100}\right)^{i-1} R, \tag{9}$$

where we choose $R \geq 100^{m+1}(M + 1)$. This in turn implies that the distance between the closest pair of true centers is at least $10^4 \times (M + 1)$.

Our random initialization strategy samples the initial centers $\mu_1, \mu_2, \cdots, \mu_M$ i.i.d. from the distribution $\mathrm{GMM}(\boldsymbol{\mu}^*)$. We can view this sampling process as two separate steps:

(i) Sample an integer $Z_i$ uniformly from $[M]$.

(ii) Sample value $\mu_i$ from the Gaussian distribution $\mathcal{N}(\mu_{Z_i}^*, \mathrm{I})$.

Concentration properties of the Gaussian distribution will ensure that $\mu_i$ will not be too far from its expectation $\mu_{Z_i}^*$. Formally, we define the following event:

$$\mathcal{E}_M := \left\{\text{all } M \text{ initial points } \mu_i \text{ are contained in } \mathbb{B}_{\mu_{Z_i}^*}(M)\right\}. \tag{10}$$

By standard Gaussian tail bounds, we have

$$\mathbb{P}(\mathcal{E}_M) = (1 - \mathbb{P}_{X \sim \mathcal{N}(0,1)}(|X| > M))^M \geq (1 - 2Me^{-M^2/2}) \geq 1 - e^{-\Omega(M)}$$

This implies that the event $\mathcal{E}_M$ will hold with high probability (when $M$ is large). Conditioned on the event $\mathcal{E}_M$, we are guaranteed that all initialized points are relatively close to some true center.

A key observation regarding the configuration of centers in the model GMM($\boldsymbol{\mu}^*$) specified by equation (9) is that the true centers can be partitioned into two well separated regions. More precisely, it is easy to verify that there are $M/2$ true centers in the interval $\mathbb{B}_{-R}(R/99)$ while the remaining $M/2$ true centers are contained in the interval $\mathbb{B}_R(R/99)$. In what follows, we refer to $\mathbb{B}_{-R}(2R/99)$ as the *left urn* and to $\mathbb{B}_R(2R/99)$ as the *right urn*.

Conditioned on $\mathcal{E}_M$, each initial point lands in either the left urn or the right urn with equal probability. Suppose we initialize EM with $(M_1, M_2)$ centers in the left and right urn respectively. By Lemma 1(a), the only three possible values of the pair $(M_1, M_2)$ for which the EM algorithm might converge to a global optimum are $(0, M), (M, 0), (M/2, M/2)$. A simple calculation will show that the first and second possibilities occur with exponentially small probability. However, the third possibility occurs with only polynomially small probability, and so we need to further investigate this possibility.

Consider, for example, the left urn: the true centers in the left urn can further be partitioned into two intervals $\mathbb{B}_{-1.01R}(R/9900)$ and $\mathbb{B}_{-0.99R}(R/9900)$ with $M/4$ true centers in each. Thus, each urn can be further partitioned into a left urn and a right urn. Following the same analysis as above and now using part (b) of Lemma 1 instead of part (a), we see that in order to ensure that the EM algorithm converges to a global optimum, the number of initial centers in $\mathbb{B}_{-1.01R}(2R/9900)$ and $\mathbb{B}_{-0.99R}(2R/9900)$ must be one of the following pairs $\{(0, M/2), (M/2, 0), (M/4, M/4)\}$.

Our configuration of centers in equation (9) guarantees that this argument can be recursively applied until we reach an interval which contains only two true centers. For a configuration of $M$ initial centers, we call these initial centers a *good initialization* for a collection of true centers if one of the following holds:

(a) $M = 1$,

(b) the number of initial centers assigned to the left urn and the right urn of the collection of true centers are either $(0, M)$ or $(M, 0)$,

(c) the number of initial centers assigned to the left urn and the right urn of the collection of true centers are $(M/2, M/2)$; and further recursively the initialization in both the left and the right urns are good initializations.

Lemma 1 implies that the EM algorithm converges to a global maximum only if a good initialization is realized. We will now show that the probability of a good initialization is exponentially small.

Let $\mathcal{F}_M$ represent the event that a good initialization is generated on a mixture with $M$ components, for our configuration of true centers. Let $M_1$ and $M_2$ represent the number of initial centers in the left urn and the right urn, respectively. Conditioning on the event $\mathcal{E}_M$ from equation (10), we have

$$\mathbb{P}(\mathcal{F}_M \mid \mathcal{E}_M) \le \mathbb{P}(M_1 = 0) + \mathbb{P}(M_1 = M) + \mathbb{P}(M_1 = M/2) \cdot \left(\mathbb{P}(\mathcal{F}_{M/2} \mid \mathcal{E}_M)\right)^2$$

$$\le 2 \times \binom{M}{0} \frac{1}{2^M} + \binom{M}{M/2} \frac{1}{2^M} \cdot \left(\mathbb{P}(\mathcal{F}_{M/2} \mid \mathcal{E}_M)\right)^2$$

$$\le \frac{1}{2^{M-1}} + \frac{1}{2} \cdot \left(\mathbb{P}(\mathcal{F}_{M/2} \mid \mathcal{E}_M)\right)^2 .$$

Since $\mathbb{P}(\mathcal{F}_1 \mid \mathcal{E}_M) = 1$, solving this recursive inequality implies that $\mathbb{P}(\mathcal{F}_M \mid \mathcal{E}_M) \le e^{-cM}$ for some universal constant $c$. Thus, the probability that the EM algorithm converges to a global maximum is upper bounded by:

$$\mathbb{P}(\mathcal{F}_M) \le \mathbb{P}(\mathcal{E}_M)\mathbb{P}(\mathcal{F}_M \mid \mathcal{E}_M) + \mathbb{P}(\overline{\mathcal{E}_M}) \le \mathbb{P}(\mathcal{F}_M \mid \mathcal{E}_M) + \mathbb{P}(\overline{\mathcal{E}_M}) \le e^{-\Omega(M)}.$$

To complete the proof for the case when $M = 2^m$ for a positive integer $m$, we need to argue that on the event $\mathbb{P}(\overline{\mathcal{F}_M})$, the log-likelihood of the solution reached by the EM algorithm can be arbitrarily worse than that of the global maximum. We claim that when the event $\overline{\mathcal{F}_M}$ occurs, the EM algorithm returns a solution $\boldsymbol{\mu}$ for which there is at least one urn containing two true centers which is assigned a single center by the EM algorithm at every iteration $t \ge 0$. As a consequence, there is at least one true center $\mu_j^*$ for which we have that $|\mu_j^* - \mu_i'| \ge \frac{R}{100^m}$ for all $i = 1, \dots, M$. Now, we claim that we can choose $R$ to be large enough to ensure an arbitrarily large gap in the likelihood of the EM solution and the global maximum. Concretely, as $R \to \infty$, we have:

$$\lim_{R \to +\infty} \mathcal{L}(\boldsymbol{\mu}) \le \lim_{R \to +\infty} \frac{1}{M} \mathbb{E}_{X \sim \mathcal{N}(\mu_j^*, 1)} \log \left(\sum_{i=1}^M e^{-\frac{1}{2}(X - \mu_i)^2}\right) - \frac{1}{M} \log(M\sqrt{2\pi}) = -\infty.$$

However, the global maximum $\boldsymbol{\mu}^*$ has log-likelihood

$$\lim_{R \to +\infty} \mathcal{L}(\boldsymbol{\mu}^*) = -\frac{1}{2} - \log(M\sqrt{2\pi}).$$

Once again we can use the continuity of the log-likelihood as a function of $R$ to conclude that there is a finite sufficiently large $R > 0$ such that the conclusion of Theorem 2 holds.

**Case $2^{m-1} < M \le 2^m$:** At a high level, we deal with this case by constructing a configuration with $2^m$ centers and pruning this down to have $M$ centers, while ensuring that the resulting urns are still approximately balanced which in turn ensures that our previous calculations continue to hold.

Our configuration of true centers in equation (9) can be viewed as the $2^m$ leaves of a binary tree with depth $M$, where the vectors $\boldsymbol{\epsilon}$ indexing the true centers represent the unique path from the root and to the leaf: the value of $\epsilon_i$ indicates whether to go down to the left child or to the right child at the $i$-th level of the tree. We choose $M$ true centers from the $2^m$ leaves by the following procedure. Starting from the root, we assign $\lceil M/2 \rceil$ true centers to the left sub-tree, and assign $\lfloor M/2 \rfloor$ true centers to the right sub-tree. For any sub-tree, suppose that it was assigned $l$ true centers, then we assign $\lceil l/2 \rceil$ true centers to its left subtree and $\lfloor l/2 \rfloor$ true centers to its right subtree. This procedure is recursively continued until all the true centers are assigned to leaves. Each leaf corresponds to a point on the real line and we choose this point as the location of the corresponding center.

The above construction has the following two properties: first, the locations of the true centers satisfy the separation requirements we used in dealing with the case when $M = 2^m$, and further the assignment of the centers to the left and right urns in each case is roughly balanced. By leveraging these two properties we can follow essentially the same steps as we did in the case with $M = 2^m$, and we omit these remaining proof details here.

## A.3 Proof of Theorem 3

We now embark on the proof of Theorem 3. The proof follows from a similar outline to the proof of Theorem 2 and we only develop the main ideas here. Concretely, it is easy to verify that in order to prove the result we only need to establish the analogue of Lemma 1 for the first-order EM algorithm.

Intuitively, we first argue that the first-order EM updates can be viewed as less aggressive versions of the corresponding EM updates, and we use this fact to argue that Lemma 1 continues to hold for the first-order EM algorithm. Concretely, we can compare the update of EM algorithm:

$$\mu_i^{\text{new, EM}} = \frac{\mathbb{E}_{\boldsymbol{\mu}^*} w_i(X) \cdot X}{\mathbb{E}_{\boldsymbol{\mu}^*} w_i(X)}$$

with the update of the first-order EM algorithm:

$$\mu_i^{\text{new, first-order EM}} = \mu_i + s\mathbb{E}_{\boldsymbol{\mu}^*} w_i(X)(X - \mu_i).$$

If for any parameter $\mu_i$, we choose the stepsize $s = \frac{1}{\mathbb{E}_{\boldsymbol{\mu}^*} w_i(X)}$, for the first-order EM algorithm, then the two updates will match for that parameter. For the first-order EM algorithm, we always use a step size $s \in (0, 1)$, while $\frac{1}{\mathbb{E}_{\boldsymbol{\mu}^*} w_i(X)} \ge 1$. Consequently, there must exist some $\theta_i \in [0, 1]$ such that

$$\mu_i^{\text{new, first-order EM}} = \theta_i \mu_i + (1 - \theta_i)\mu_i^{\text{new, EM}}.$$

Thus, we see that the first-order EM update is a less aggressive version of the EM update. An examination of the proof of Lemma 1 reveals that this property suffices to ensure that its guarantees continue to hold for the first-order EM algorithm, which completes the proof of Theorem 3.

## A.4 Proof of Theorem 4

In this section, we prove Theorem 4. Throughout this proof, we use the fact that the first-order EM updates with step size $s \in (0, 1)$ take the form

$$\boldsymbol{\mu}^{\text{new}} = \boldsymbol{\mu} + s\nabla\mathcal{L}(\boldsymbol{\mu}). \tag{11}$$

In order to reason about the behavior of the first-order EM algorithm, we first provide a result that concerns the Hessian of the log-likelihood.

**Lemma 2.** *For any scalar $s \in (0, 1)$ and for any $\boldsymbol{\mu}$, we have $s\nabla^2\mathcal{L}(\boldsymbol{\mu}) \succ -I$.*

We prove this claim at the end of the section. Taking this lemma as given, we can now prove the theorem's claims. We first show that the first-order EM algorithm with stepsize $s \in (0, 1)$ converges to a critical point. By a Taylor expansion of the log-likelihood function, we have

$$\mathcal{L}(\boldsymbol{\mu}^{\text{new}}) = \mathcal{L}(\boldsymbol{\mu}) + \langle \nabla\mathcal{L}(\boldsymbol{\mu}), \boldsymbol{\mu}^{\text{new}} - \boldsymbol{\mu} \rangle + \frac{1}{2}(\boldsymbol{\mu}^{\text{new}} - \boldsymbol{\mu})^T \nabla^2\mathcal{L}(\widetilde{\boldsymbol{\mu}})(\boldsymbol{\mu}^{\text{new}} - \boldsymbol{\mu}),$$

for some $\widetilde{\boldsymbol{\mu}}$ on the line joining $\boldsymbol{\mu}$ and $\boldsymbol{\mu}^{\text{new}}$. Applying Lemma 2 guarantees that

$$\mathcal{L}(\boldsymbol{\mu}^{\text{new}}) \geq \mathcal{L}(\boldsymbol{\mu}) + \langle \nabla\mathcal{L}(\boldsymbol{\mu}), \boldsymbol{\mu}^{\text{new}} - \boldsymbol{\mu} \rangle - \frac{1}{2s}\|\boldsymbol{\mu}^{\text{new}} - \boldsymbol{\mu}\|_2^2.$$

From the form (11) of the gradient EM updates, we then have

$$\mathcal{L}(\boldsymbol{\mu}^{\text{new}}) \geq \mathcal{L}(\boldsymbol{\mu}) + \left(s - \frac{s}{2}\right)\|\nabla\mathcal{L}(\boldsymbol{\mu})\|_2^2.$$

Consequently, for any choice of step size $s \in (0, 1)$ and any point $\boldsymbol{\mu}$ for which $\nabla\mathcal{L}(\boldsymbol{\mu}) \neq 0$, applying the gradient EM update leads to a strict increase in the value of the population likelihood $\mathcal{L}$. Since $\mathcal{L}$ is upper bounded by a constant for a mixture of $M$ spherical Gaussians, we can conclude that first-order EM must converge to some point. It is easy to further verify that it must converge to a point for which $\nabla\mathcal{L}(\boldsymbol{\mu}) = 0$ which concludes the first part of our proof.

Next we show that the first-order EM algorithm will not converge to strict saddle points almost surely. We do this via a technique that has been used in recent papers [Lee et al., 2016, Panageas and Piliouras, 2016], exploiting the stable manifold theorem from dynamical systems theory. For this portion of the proof, it will be convenient to view the first-order EM updates as a map from the parameter space to the parameter space; i.e., we define the first-order EM map by:

$$\mathbf{g}(\boldsymbol{\mu}) := \boldsymbol{\mu} + s\nabla\mathcal{L}(\boldsymbol{\mu}). \tag{12}$$

Recalling Definition 1 of strict saddle points, we denote by $\mathcal{D}_{ss}$ the set of initial points from which the first-order EM algorithm converges to a strict saddle point. With these definitions in place, we can state an intermediate result:

**Lemma 3** ([Lee et al., 2016, Panageas and Piliouras, 2016]). *If the map $\boldsymbol{\mu} \mapsto \mathbf{g}(\boldsymbol{\mu})$ defined by equation (12) is a local diffeomorphism for each $\boldsymbol{\mu}$, then $\mathcal{D}_{ss}$ has zero Lebesgue measure.*

Denote the Jacobian matrix of map $\mathbf{g}$ at point $\mu$ as $\nabla\mathbf{g}(\boldsymbol{\mu})$ where $[\nabla\mathbf{g}(\boldsymbol{\mu})]_{ij} = \frac{\partial}{\partial\mu_j}g_i(\boldsymbol{\mu})$. By Lemma 2, the Jacobian $\nabla\mathbf{g}(\boldsymbol{\mu}) = \mathrm{I} + s\nabla^2\mathcal{L}(\boldsymbol{\mu})$ is strictly positive definite, and hence invertible for all $\boldsymbol{\mu}$, which implies that the map $\mathbf{g}$ is a local diffeomorphism everywhere. Furthermore, our random initialization strategy specifies the distribution of the initial point $\boldsymbol{\mu}^{(0)}$ which is absolutely continuous with respect to Lebesgue measure. Combined these facts with lemma 3, we have proved Theorem 4.

Finally, the only remaining detail is to prove Lemma 2. By definition, we have

$$\mathrm{I} + s\nabla^2\mathcal{L}(\boldsymbol{\mu}) = \underbrace{\begin{bmatrix} (1 - s\mathbb{E}w_1(X))\mathrm{I}_d & \cdots & 0 \\ & \cdots & \cdots \\ 0 & \cdots & (1 - s\mathbb{E}w_M(X))\mathrm{I}_d \end{bmatrix}}_{:= \mathbf{D}} + s\mathbf{Q},$$

where the matrix $\mathbf{Q}$ has $d$-dimensional blocks of the form

$$\mathbf{Q}_{ij} = \begin{cases} \mathbb{E}(w_i(X) - w_i^2(X))(X - \mu_i)(X - \mu_i)^\top & \text{if } i = j \\ -\mathbb{E}w_i(X)w_j(X)(X - \mu_i)(X - \mu_j)^\top & \text{otherwise.} \end{cases}$$

Since $w_i(X) \leq 1$ for all $i \in [M]$ and $s < 1$, it follows that the diagonal matrix $\mathbf{D}$ is strictly positive definite. Consequently, in order to prove Lemma 2, it suffices to show that $\mathbf{Q}$ is positive semidefinite. Letting $\mathbf{v} = (\mathbf{v}_1^\top, \ldots, \mathbf{v}_M^\top)^\top$, where $\mathbf{v}_i \in \mathbb{R}^d$, be arbitrary vectors, we have:

$$\mathbf{v}^\top\mathbf{Q}\mathbf{v} = \sum_{i=1}^M \mathbb{E}w_i(X)[\mathbf{v}_i^\top(X - \mu_i)]^2 - \sum_{i=1}^M\sum_{j=1}^M \mathbb{E}w_i(X)w_j(X)[\mathbf{v}_i^\top(X - \mu_i)][\mathbf{v}_j^\top(X - \mu_j)]$$

$$\overset{(i)}{\geq} \sum_{i=1}^M \mathbb{E}w_i(X)[\mathbf{v}_i^\top(X - \mu_i)]^2 - \sum_{i=1}^M\sum_{j=1}^M \frac{1}{2}\left[\mathbb{E}w_i(X)w_j(X)[\mathbf{v}_i^\top(X - \mu_i)]^2 + \mathbb{E}w_i(X)w_j(X)[\mathbf{v}_j^\top(X - \mu_j)]^2\right]$$

$$\overset{(ii)}{=} \sum_{i=1}^M \mathbb{E}w_i(X)[\mathbf{v}_i^\top(X - \mu_i)]^2 - \sum_{i=1}^M \mathbb{E}w_i(X)[\mathbf{v}_i^\top(X - \mu_i)]^2 = 0,$$

where step (i) uses the elementary inequality $|ab| \leq \frac{1}{2}(a^2 + b^2)$; and step (ii) uses the fact that $\sum_{i=1}^{M} w_i(X) = 1$ for any $X$. This completes the proof.

# B  Proofs of Technical Lemmas

The bulk of this section is devoted to the proof of Lemma 1, which is based on a number of technical lemmas.

## B.1  Proof of Lemma 1

Underlying our proof is the following auxiliary result:

**Lemma 4.** *Suppose that:*

(a) *The true distribution is a GMM($\boldsymbol{\mu}^*$) with $M$ components and that all true centers are located in $(-\infty, -10a) \cup (a, +\infty)$ with at least one center in $(a, 3a)$, with $a > \log M + 3$.*

(b) *The current configuration of centers has the property that for any true center $\mu_j^*$ in $(-\infty, -10a)$, there exists a current center $\mu_{j'}$ such that $|\mu_{j'} - \mu_j^*| \leq |\mu_j^*|/6$.*

*Then, for any $i \in [M]$ for which the current parameter $\mu_i \in [0, 4a]$, we have $\mathbb{E}w_i(X)X \geq 0$.*

See Section B.2 for the proof of this claim.

Using Lemma 4, let us now prove Lemma 1. Without loss of generality, we may assume that $\mu_i \in \mathbb{B}_{c\delta}(2\delta)$, for some $i \in [M]$. Thus, in order to establish the claim, it suffices to show that after one step of the EM algorithm, the new iterate $\mu_i^{\text{new}}$ belongs to $\mathbb{B}_{c\delta}(2\delta)$ as well.

In order to show that $\mu_i^{\text{new}} \in \mathbb{B}_{c\delta}(2\delta)$, note that by the update equation (5), we have $\mu_i^{\text{new}} = \frac{\mathbb{E}w_i(X)X}{\mathbb{E}w_i(X)}$. Thus, it is equivalent to prove that

$$\mathbb{E}w_i(X)(X - (c-2)\delta) \geq 0, \quad \text{and} \quad \mathbb{E}w_i(X)(X - (c+2)\delta) \leq 0.$$

The first inequality can be proved by substituting $Z = X - (c-2)\delta$ and applying Lemma 4 to $Z$. Similarly, the second inequality can be proved by defining $Y := (c+2)\delta - X$, and then applying Lemma 4 to $Y$.

## B.2  Proof of Lemma 4

Our proof of this claim hinges on two auxiliary lemmas, which we begin by stating. Intuitively, our first lemma shows that if the data are generated by a single Gaussian, whose mean is at least $\Omega(\log M)$ to the right of the origin, then it will affect any $\mu_i \geq 0$, by forcing it to the right no matter where the other $\{\mu_j\}_{j \neq i}$ are.

**Lemma 5.** *Suppose that the true distribution is a unit variance Gaussian with mean $\mu^* \geq a$ for some $a > \log M + 3$, and that the current configuration of centers, $\mu_1, \cdots, \mu_M$, has the $i^{\text{th}}$ center $\mu_i \geq 0$. Then we have*

$$\mathbb{E}w_i(X)X \geq 0. \tag{13a}$$

*Furthermore, if $\mu^* \leq 3a$, and $0 \leq \mu_i \leq 4a$, then:*

$$\mathbb{E}w_i(X)X \geq \frac{a}{5M}e^{-9a^2/2}. \tag{13b}$$

See Section B.3 for the proof of this claim. In a similar vein, if the data is generated by a single Gaussian far to the left of the origin, and some current center $\mu_j$ is sufficiently close to it then this Gaussian will force $\mu_i$ towards the negative direction, but will only have a small effect on $\mu_i$. More formally, we have the following result:

**Lemma 6.** *Suppose that the true distribution is a unit variance Gaussian with mean $\mu^* = -r$, and that the current configuration of centers, $\mu_1, \cdots, \mu_M$, has the $i^{\text{th}}$ center $\mu_i \geq 0$ and further has at least one $\mu_j$ such that $|\mu_j - \mu^*| \leq \frac{r}{6}$. Then we have that:*

$$\mathbb{E}w_i(X)X \geq -3re^{-r^2/18}. \tag{14}$$

See Section B.4 for the proof of this claim.

Equipped with these two auxiliary results, we can now prove Lemma 4. Without loss of generality, suppose that the centers are sorted in ascending order, and that the $\ell^{\text{th}}$ true center is the smallest true center in $(0, +\infty)$. From the assumptions of Lemma 4, we know $\mu_\ell^*$ belongs to the interval $(a, 3a)$. Thus, when $X$ is drawn from a Gaussian mixture, we have

$$\mathbb{E}w_i(X)X = \frac{1}{M} \sum_{j=1}^{M} \mathbb{E}_{X \sim \mathcal{N}(\mu_j^*, 1)} w_i(X)X$$

$$= \frac{1}{M} \sum_{j=1}^{\ell-1} \mathbb{E}_{X \sim \mathcal{N}(\mu_j^*, 1)} w_i(X)X + \frac{1}{M} \mathbb{E}_{X \sim \mathcal{N}(\mu_\ell^*, 1)} w_i(X)X + \frac{1}{M} \sum_{j=\ell+1}^{M} \mathbb{E}_{X \sim \mathcal{N}(\mu_j^*, 1)} w_i(X)X.$$

We now use Lemma 6 to bound the first term. Since $f(y) = -3y \cdot e^{-y^2/18}$ is monotonically increasing in $[3, +\infty)$, and from the assumptions of Lemma 4, we have $|\mu_j^*| > -10a > -(9a+2)$ for all $j < \ell$. Then:

$$\frac{1}{M} \sum_{j=1}^{\ell-1} \mathbb{E}_{X \sim \mathcal{N}(\mu_j^*, \frac{1}{2})} w_i(X)X \geq -\frac{1}{M} \sum_{j=1}^{\ell-1} 3 \mid \mu_i \mid e^{-\mu_i^2/18} \geq -3(9a+2)e^{-(9a+2)^2/18}.$$

By Lemma 5, we know that the third term is non-negative and that the second term can be lower bounded by a sufficiently large quantity. Putting together the pieces, we find that

$$\mathbb{E}_X w_i(X)X \geq -3(9a+2)e^{-9a^2/2 - 2a - \frac{2}{9}} + \frac{a}{5M^2} e^{-9a^2/2}$$

$$\geq e^{-9a^2/2} \left[ \frac{a}{5M^2} - 3(9a+2)e^{-2\log M - 6} \right]$$

$$\geq \frac{e^{-9a^2/2 - 6}}{M^2} [80a - 3(9a+2)]$$

$$\geq 0,$$

which completes the proof.

## B.3 Proof of Lemma 5

Introducing the shorthand $w^* := \min_{x \in [1,2]} w_i(x)$, we have

$$\mathbb{E}w_i(X)X \geq \frac{1}{\sqrt{2\pi}} \int_{-\infty}^{0} w_i(x)xe^{-(x-\mu^*)^2/2} \mathrm{d}x + \frac{1}{\sqrt{2\pi}} \int_{1}^{2} w_i(x)xe^{-(x-\mu^*)^2/2} \mathrm{d}x$$

$$+ \frac{1}{\sqrt{2\pi}} \int_{a}^{3a} w_i(x)xe^{-(x-\mu^*)^2/2} \mathrm{d}x.$$

We calculate the first two terms: for this purpose, the following lemma is useful:

**Lemma 7.** *For any* $\mu_1, \cdots, \mu_M$ *where* $\mu_i \geq 0$, *we have following:*

$$\min_{x \in [1,2]} w_i(x) \geq \frac{1}{Me^2} \max_{x \in (-\infty, 0]} w_i(x). \tag{15}$$

See Section B.3.1 for the proof of this claim. From Lemma 7, we have that:

$$\frac{1}{\sqrt{2\pi}} \int_{-\infty}^{0} w_i(x)xe^{-(x-\mu^*)^2/2} \mathrm{d}x + \frac{1}{\sqrt{2\pi}} \int_{1}^{2} w_i(x)xe^{-(x-\mu^*)^2/2}$$

$$\geq \frac{1}{\sqrt{2\pi}} \left[ \int_{-\infty}^{0} Me^2 w^* xe^{-(x-\mu^*)^2/2} \mathrm{d}x + \int_{1}^{2} w^* xe^{-(x-\mu^*)^2/2} \mathrm{d}x \right]$$

$$\geq \frac{w^*}{\sqrt{2\pi}} \left[ \int_{-\infty}^{0} Me^2 (x-\mu^*) e^{-(x-\mu^*)^2/2} \mathrm{d}x + \int_{1}^{2} xe^{-(x-\mu^*)^2/2} \mathrm{d}x \right]$$

$$\geq \frac{w^*}{\sqrt{2\pi}} \left[ -Me^{2-(\mu^*)^2/2} + e^{-(\mu^*-1)^2/2} \right] = \frac{w^* e^{-(\mu^*)^2}}{\sqrt{2\pi}} \left[ e^{\mu^*-1/2} - Me^2 \right] \geq 0.$$

The last inequality holds since $\mu^* > a > \log M + 3$. The third term is always positive, and this finishes the proof of first claim.

For second claim: if we further know that $\mu^* \leq 3a$, and $\mu_i \leq 4a$, then for any $x \in [a, 3a]$, $w_i(x) \geq \frac{e^{-9a^2/2}}{M}$, we have:

$$\frac{1}{\sqrt{2\pi}} \int_a^{3a} w_i(x) x e^{-(x-\mu^*)^2/2} \mathrm{d}x \geq \frac{1}{M\sqrt{2\pi}} e^{-9a^2/2} a \int_a^{3a} e^{-(x-\mu^*)^2/2} \mathrm{d}x$$
$$\geq \frac{a}{M\sqrt{2e\pi}} e^{-9a^2/2} \geq \frac{a}{5M} e^{-9a^2/2}.$$

The last inequality is true by integrating over an interval of length 1 around $\mu^*$ contained in $(a, 3a)$.

### B.3.1 Proof of Lemma 7

We split the proof into two cases.

**Case $\mu_i \in [0, 2]$:** In this case, we are guaranteed that $\max_{x \in (-\infty, 0]} w_i(x) \leq 1$. Also, for any $x \in [1, 2]$, we have:

$$w_i(x) = \frac{e^{-(x-\mu_i)^2/2}}{\sum_j e^{-(x-\mu_j)^2/2}} \geq \frac{1}{Me^2}, \tag{16}$$

which proves the required result.

**Case $\mu_i > 2$:** In this case, we have

$$w_i(x) = \frac{e^{-(x-\mu_i)^2/2}}{\sum_j e^{-(x-\mu_j)^2/2}} = \frac{1}{\sum_{j \neq i} \frac{1}{M-1} + e^{[(x-\mu_i)^2 - (x-\mu_j)^2]/2}}$$
$$= \frac{1}{\sum_{j \neq i} \frac{1}{M-1} + e^{(\mu_i - \mu_j)(\mu_i + \mu_j - 2x)/2}} = \frac{1}{\sum_{j \neq i} A_{ij}(x)},$$

where $A_{ij}(x) := \frac{1}{M-1} + e^{(\mu_i - \mu_j)(\mu_i + \mu_j - 2x)/2}$. It suffices to show that

$$A_{ij}(x) \leq M A_{ij}(x') \quad \text{for any } x \in [1, 2], x' \in (-\infty, 0] \text{ and } j \in [M]. \tag{17}$$

Using this, we know:

$$w_i(x) = \frac{1}{\sum_{j \neq i} A_{ij}(x)} \geq \frac{1}{\sum_{j \neq i} M A_{ij}(x')} = \frac{1}{M} w_i(x'), \tag{18}$$

and the claim of Lemma 7 easily follows. In order to establish the claim of equation (17), we note that if $\mu_j \leq \mu_i$, then since $x' < x$, we have

$$(\mu_i - \mu_j)(\mu_i + \mu_j - 2x) \leq (\mu_i - \mu_j)(\mu_i + \mu_j - 2x'),$$

which implies that $A_{ij}(x) \leq A_{ij}(x')$. If $\mu_i < \mu_j$, then we know:

$$(\mu_i - \mu_j)(\mu_i + \mu_j - 2x) < 0. \tag{19}$$

This implies $A_{ij}(x) \leq \frac{1}{M-1} + 1 = \frac{M}{M-1}$. On the other hand, we always have $A_{ij}(x') \geq \frac{1}{M-1}$, this gives $A_{ij}(x) \leq M A_{ij}(x')$, which finishes the proof.

### B.4 Proof of Lemma 6

We have

$$\mathbb{E}w_i(X)X \geq \frac{1}{\sqrt{2\pi}} \int_{-\infty}^{-2r/3} w_i(x) x e^{-(x-\mu^*)^2/2} \mathrm{d}x + \frac{1}{\sqrt{2\pi}} \int_{-2r/3}^0 w_i(x) x e^{-(x-\mu^*)^2/2} \mathrm{d}x.$$

For the first term, we know for any $x \in (-\infty, -2r/3]$, we have:

$$w_i(x) \leq \frac{e^{-x^2/2}}{e^{-(x-\mu_j)^2/2}} = e^{-x\mu_j + \mu_j^2/2} \leq e^{-\frac{2r}{3}\mu_j + \mu_j^2/2} \leq e^{-\frac{7}{72}r^2} \leq e^{-r^2/18}.$$

The second last inequality is true since $\mu_j \geq -\frac{7r}{6}$. Thus, we know:

$$
\frac{1}{\sqrt{2\pi}} \int_{-\infty}^{-2r/3} w_i(x)xe^{-(x-\mu^*)^2/2}\mathrm{d}x \geq \frac{e^{-r^2/18}}{\sqrt{2\pi}} \int_{-\infty}^{-2r/3} xe^{-(x-\mu^*)^2/2}\mathrm{d}x
$$

$$
\geq \frac{e^{-r^2/18}}{\sqrt{2\pi}} \left[ \int_{-\infty}^{-2r/3} (x-\mu^*)e^{-(x-\mu^*)^2/2}\mathrm{d}x + \mu^*\sqrt{2\pi} \right]
$$

$$
\geq \frac{e^{-r^2/18}}{\sqrt{2\pi}} \left[ -\frac{1}{2}e^{-r^2/18} - r\sqrt{2\pi} \right] \geq -2re^{-r^2/18}.
$$

For the second term, we have:

$$
\frac{1}{\sqrt{2\pi}} \int_{-2r/3}^{0} w_i(x)xe^{-(x-\mu^*)^2/2}\mathrm{d}x \geq -\frac{2r}{3\sqrt{2\pi}} \int_{-2r/3}^{0} e^{-(x-\mu^*)^2/2}\mathrm{d}x
$$

$$
\geq -\frac{2r}{3\sqrt{2\pi}} \int_{-2r/3}^{+\infty} e^{-(x-\mu^*)^2/2}\mathrm{d}x \geq -\frac{2r}{3}e^{-r^2/18}.
$$

Putting the pieces together we obtain,

$$
\mathbb{E}w_i(X)X \geq -(\frac{2}{3}+2)e^{-r^2/18} \geq -3re^{-r^2/18},
$$

as desired.