[Reviews · NeurIPS 2016]

Reviewer 1

Summary

The paper studies the convergence of the EM algorithm (and iterative heuristics for Maximum Likelihood Estimation) for learning mixtures of spherical Gaussians, and presents the following results that show non-convergence to global maxima: 1. Even when the mixture has 3 clusters that are very far apart and with infinite samples, algorithms like EM can get stuck in local optima. 2. For mixtures of k gaussians, random initialization for EM only succeeds with at most exp(-\Omega(k)) probability. 3. Gradient EM will not converge to strict saddle points generically; hence, bad local maxima are typically the issue.

Qualitative Assessment

The paper presents a coherent set of results that show that EM does not converge to global maxima (even with infinite samples). However, I feel the results are not very surprising, and the techniques also follow expected lines. The main bad example is a mixture with 3 components: two of them are closer to each other, and the third is very far from both. They show that for the log-likelihood objective, there is a local maxima with one center near the first two components, and two centers near the far-off cluster. This is fairly expected behavior, particularly when the centers are separated by >> \sqrt{d}. Under this amount of separation, the clusters do not overlap and learning mixtures of spherical Gaussians becomes very similar in flavor to k-means. Similar examples provide bad instances of Lloyd's algorithm for k-means clustering. This example is extended for k components with a nice recursive construction -- this requires some effort, but again somewhat expected. Similarly, random initialization fails for similar reasons as for k-means clustering with the same failure probability (for k-means this serves as the motivation to pick k logk centers, or distance-squared sampling). To conclude, while the results tell a coherent story about the non-convergence to global optima, I feel this is along somewhat expected lines. Comments: 1. Kumar-Kannan provides convergence guarantees for Lloyd's heuristic for k-means clustering (and hence mixtures of Gaussians under sufficient separations conditions). While EM is different from Lloyd's heuristic when the Gaussians are not sufficiently separated (as in Balakrishnan-Wainwright-Yu), they seem essentially when there is large separation (this is true of the bad examples here). So, I think a comparison with k-means (and Lloyd's heuristic) that also contrasts this result would be very useful.

Confidence in this Review

2-Confident (read it all; understood it all reasonably well)


Reviewer 2

Summary

This paper addresses the expectation-maximization (EM) algorithm for the estimation of the set of mean parameters for a model of mixture of Gaussian variables. The number of Gaussian variables, the weights of the mixture, and the covariance matrices of the Gaussian variables are known and fixed. The paper is theoretical and prove three negative results for the EM algorithm. 1) There can be, even for large sample size, local maxima for the likelihood function, with likelihood values arbitrary smaller than the maximal likelihood value. 2) The EM algorithm, randomly initiated, converges with high probability to a critical point with likelihood value arbitrarily smaller than the maximal likelihood value. 3) The same negative result holds for the gradient EM algorithm.

Qualitative Assessment

I think that the question addressed by the paper is interesting and worthy of attention. In order to obtain rigorous proofs, the paper is restricted to a simplified mixture of Gaussian variable model: the number of components is known, the weights are known and uniform and the covariance matrices are equal to identity. I do not think that these simplifications are a problem, given that the proofs are already technical, and that the obtained results are already informative. These results are very interesting in my opinion, and can be useful to understand better the EM algorithm in practice. It should be noted that the paper is written as addressing a case of general dimension d, while the proofs address the case d=1. Perhaps the authors could write if similar proofs would be possible for arbitrary dimension d, and if not, mention briefly the additional difficulties of the case d larger than 1. My only concern about the paper is the proof of Theorem 1 (unfortunately, I did not read the proofs in the supplementary material). I think that this proof is too short, and I would have preferred it to be longer, at the expense of shortening some of the discussions in the paper. The part of the proof where I think more details should be given is from line 288 to 306. The authors provide limits (as gamma or R goes to infinity) of supremums of likelihood functions over some domains. I do not have a problem with the computation of limits of likelihood functions evaluated a fixed parameters, so that, for instance, I conceive that the three equations of line 288 should be correct. Nevertheless, obtaining the limit of a maximum is not entirely obvious to me, when the domain is unbounded or depend of the parameters that go to infinity. In the same way, the authors mention at some places that the likelihood function is continuous, but this does not imply directly the types of convergences stated in the paper, in my opinion. I think that more details, and so a longer proof, are needed so that the reader can be completely convinced of the validity of the proof of Theorem 1. Finally, I have an other minor concern with the part of the paper addressing the Gradient EM, from line 162 to line 172. I found this part more difficult to follow than the rest of the paper. In particular, perhaps the authors could give more explanation on how Equation (5) is obtained.

Confidence in this Review

1-Less confident (might not have understood significant parts)


Reviewer 3

Summary

The paper is concerned with the existence of local modes in the log-likelihood of Gaussian mixture models. The simplest possible mixture model is considered: isotropic Gaussian components with known scale. In addition the model is well-specified. The authors show that local modes exist, even when infinite data is available, and that these are arbitrarily bad, in term of differences in log-likelihood between the local and global optima. The paper focuses on particular configurations, in particular cases where the mixture centres are divided between well-spaced clusters. Under such configurations, the authors show that random initialization will make so that EM misses the global optima almost certainly, as the number of components increases.

Qualitative Assessment

The paper is well written, with occasional typos. I have two main doubts about the practical relevance of the paper. 1. Theorem 1 shows that the log-likelihood has a local modes, even when infinite data is available. The authors show that this mode exists when a particular configuration of the centres is chosen. My question is: how likely is this configuration to occur in practice? I would feel more confident about the practical relevance of the paper if, for instance, the authors made some assumptions about the distribution of the centres, and then showed that "bad" configurations become increasingly likely as the number of components (k) or the dimensionality of the problem increases. L210 to L218 of the paper say that Theorem 2 addresses these concerns, but I am not sure it does. It would be good to have a clear statement say how likely "bad" configurations are, under some assumptions. Notice that here I am talking about "bad" configurations (positions of the true centres), not "bad" initializations. 2. In theorem 2 the initialization of the EM algorithm comes to play. The authors consider initializing the centres of the components by randomly sampling the mixture. With this initialization, and under specific configurations of the centres, the EM very often fails to converge to the global optimum. Now, let us consider the case where the data is generated using 3 Gaussian, 2 of which are close and one far apart. If we have the same number N of data-points from each Gaussian, then 2N points will fall close to the first cluster and N will fall close to the 3rd density. It seems to me that very odd that somebody could initialize EM by putting 2 centers where there are only N points and 1 center where there are N observations, especially given that the two clusters are very well separated from each other. Is random initialization practically relevant when the data is clearly clustered? L74-81 I am not sure what the example on lines 74-81 is meant to convey. It does not convince me that Sresbro conjecture holds when k = 2, so what is its purpose? L233-235 I understand requirement (2) here, but I don't see why requirement (1) is needed. You mean that, if you initialize a center between 2 clusters, you cannot guarantee that it will move toward the correct cluster? L267-273 The authors prove that EM is unlikely to converge to a saddlepoint. From a practical point of view, why is this result important? Is converging to a saddlepoint more or less problematic than converging to a local optima? MINOR POINTS: L18 "We further show gradient EM algorithm" add "that the" before "gradient". L38 "quality of the estimate" maybe "estimates" is better. L60 "to characterize when it is that efficient algorithms..." I guess here you really mean "to characterize under which conditions efficient algorithms..." L138 Just to be clear: O(d^1/2)-separated mean that all densities are c-separated, with c being O(d^1/2). L172 Isn't this sentence redundant? Please reformulate it or remove it. L177 "this is equivalent of sampling " maybe "equivalent to" is better. L218 I don't understand what a "well-separated" constant is. L220 " initialized by the the random initialization" L225 I don't know what \Omega (k) means. L245 "unfortunate coincident for one single algorithm" used "coincidence" instead. L249 "Recall for uniform weighted" add "that" after "recall" L308 "And claim when" add "that" after "claim"

Confidence in this Review

2-Confident (read it all; understood it all reasonably well)


Reviewer 4

Summary

This paper studies the structure of local maxima of the log-likelihood function of Gaussian Mixture Models (with equal mixing weights and identity as covariance). It presented three related results regarding this vein: 1. It provided a counter-example showing that there exist local maxima (when k = 3 and d=1), whose likelihood is arbitrarily worse than the global maximum. 2. They showed with random initialization, the probability of not converging to a bad local maximum could be exponentially small in k on when the means of Gaussians are configured in a special way. 3. They also showed gradient EM does not converge to a saddle point of the log-likelihood almost surely.

Qualitative Assessment

Summary: This paper studies the structure of local maxima of the log-likelihood function of Gaussian Mixture Models (with equal mixing weights and identity as covariance). It presented three related results regarding this vein: 1. It provided a counter-example showing that there exist local maxima (when k = 3 and d=1), whose likelihood is arbitrarily worse than the global maximum. 2. They showed with random initialization, the probability of not converging to a bad local maximum could be exponentially small in k on when the means of Gaussians are configured in a special way. 3. They also showed gradient EM does not converge to a saddle point of the log-likelihood almost surely. Technical quality I only checked the proofs in the main paper (Theorem 1), and I think the proofs are sound. Having read the analysis of Theorem 1, I think the proof idea for Theorem 2 also seems reasonable, although I didn’t check the details in the Appendix. All of the three theorems are well interpreted after their statements. Novelty & Originality The paper ties the mean configuration of GMM to the local maxima of its log-likelihood. I think this observation, although intuitive, is very original. And I think the key idea of Theorem 2 based on Theorem 1, that the hierarchical grouping structure of the true mean determines the local maxima for k > 3 is also very clever. Potential impact & Usefulness I think the results could have good impact on both theory and practice: in theory, it could inspire the study of the local optima structure of related clustering tasks, such as k-means problems; it could be also interesting to examine whether their result in Lemma 7 also holds for d > 1; in practice, it shows the importance of seeding enough points in each true cluster, and maybe will provide some insights in designing new algorithms. Clarity and presentation The paper does a very good job in presenting their results, and explaining the intuition of their analysis. It clearly introduces the problem setup, and provided intuition and interpretation of all their analysis.

Confidence in this Review

3-Expert (read the paper in detail, know the area, quite certain of my opinion)


Reviewer 5

Summary

The paper presents an answer to the open question asked by Srebro in his 2007 paper "Are there local maxima in the infinite-sample likelihood of Gaussian mixture estimation?". This paper answers the question in the negative through the construction of a general class of counterexamples. The authors begin be outlining the schemes by which the likelihood is maximized in the setting of Gaussian Mixture Models. These include the EM and gradient EM algorithms. The authors also address the issue of initialization and give a common choice. The authors then state their primary theorems, which give the counterexample to the question by Srebro [2007]. The authors further show even with the common choice of random initialization, the EM and gradient EM algorithms converge to the non-optimal maxima with exponentially high probability. The authors then give some intuition for their proofs by proving a simple case (k=3, d=1).

Qualitative Assessment

This paper is very clear and well written, and balances well the space requirements of the paper with the need to give intuition for their theoretical result. The question answered by the authors is of great importance for further understanding the issues that arise with non-convex optimization, even in the asymptotic limit for simple distributions. I did not carefully check the supplementary material, but some estimates of the constants in the probability term that they give would be nice. This is important because it seems to me that one would never consider k to be too large. Estimates of the constants would be nice to see when we could start to see the convergence to bad solutions in the asymptotic limit (i.e. where is the phase transition?). If the constants are too complicated or do not make sense, supporting simulations would be nice to at least this is a phenomena that we can observe. However, this issue does not detract from the novelty or impact of this paper.

Confidence in this Review

2-Confident (read it all; understood it all reasonably well)